# A Review on Research Progress in Plasma-Controlled Superwetting Surface Structure and Properties

**DOI:** 10.3390/polym14183759

**Published:** 2022-09-08

**Authors:** Dayu Li, Kai Xu, Yanjun Zhang

**Affiliations:** School of Mechanical Engineering, Yangzhou University, Yangzhou 225009, China

**Keywords:** plasma treatment, superhydrophobic, superhydrophilic, super oil repellent materials

## Abstract

Superwetting surface can be divided into (super) hydrophilic surface and (super) hydrophobic surface. There are many methods to control superwetting surface, among which plasma technology is a safe and convenient one. This paper first summarizes the plasma technologies that control the surface superwettability, then analyzes the influencing factors from the micro point of view. After that, it focuses on the plasma modification methods that change the superwetting structure on the surface of different materials, and finally, it states the specific applications of the superwetting materials. In a word, the use of plasma technology to obtain a superwetting surface has a wide application prospect.

## 1. Introduction

Superwetting surface is mainly supplied in the fields of oil–water separation [1], anti-icing [2], anti-fog [3], self-cleaning [4], packaging [5] and biology [6]. Two points may be considered for the preparation methods of a superhydrophobic surface [7]: one is to select materials with low surface energy to expand the irregularity of the material surface, the second is to reduce the energy of the rough surface and modify the rough surface with low surface energy materials. On the contrary, the superhydrophilic surface is normally obtained by forming roughness on the surface of high surface energy material. Superhydrophobic surface [8] is usually grafted with low-energy functional groups, such as nitrogen-containing and fluorine-containing functional groups. Superhydrophilic surface usually contains high-energy functional groups [9] such as hydroxyl, carboxyl, and ester groups.

Generally speaking, there are two main methods to prepare super oil repellent materials [10,11]. One type is to use fluorine-containing group materials with low surface energy to modify the membrane surface, so as to directly form micro and nano structures and expand the irregularity of the material surface. Another type is to form micro nanostructures on the surface of the membrane, and then modify the membrane surface with fluorine-containing groups with low surface energy.

Currently, many approaches have been applied to obtain superwetting surfaces [12] represented by bottom-up ones, such as immersion coating, electrospinning, self-assembly, and top-down ones, such as photolithography and the template method. The former prepared surfaces with different morphologies and the latter generated regular surface topography, both on the surface of materials [13].

Low temperature plasma technology [14] developed in recent years is a flexible and effective way to regulate the surface structure of materials, and low temperature plasma enhanced chemical vapor deposition (PECVD) technology [15,16] has certain advantages in direct surface pretreatment, low temperature modification, low organic content, and no post-processing. This technology mainly uses the energy or activity of electrons, ions, free atoms, and free radicals in the plasma to induce physical and chemical effects, cause collision, scattering, excitation, rearrangement, isomerization, defect, crystallization, and amorphization on the material surface, and finally, form new functional groups with positive functions or film layers with special structure. Therefore, the plasma technology process plays an important role in superwetting surface treatment. In 2000, there were fewer than 3500 papers published on plasma control of surface superwettability. By 2010, there were nearly 9000 papers that can be searched in this field. So far this year, there have been about 20,000 papers published in the field of plasma control of surface superwettability.

Currently, there are two widely used modification methods [17]: one is atmospheric pressure plasma [18] treatment, including dielectric barrier discharge plasma and atmospheric pressure plasma jet treatment; the other is low pressure plasma treatment [19], including radiofrequency (rf) discharge plasma treatment and glow discharge plasma treatment. The surface can be bombarded by plasma to produce nano-scale roughness [20], or the corresponding wettability of the surface can be obtained by introducing precursor grafted with various functional groups [21]. The same material can be treated by different processes, and the same process can also treat different materials. Figure 1 shows the treatment of different materials by various processes.

Hence, the main objective of this review paper is to discuss factors that affect the superwetting surface structure and performance, and study in more detail the application of plasma technology in superwetting surface structure and properties. Then, the review focuses on the current state of the plasma modification methods for different types of materials, including textile fiber surface, porous material surface, polymer film, wood surface, glass surface, and particle/powder surface. Different applications of superwetting materials are also briefly introduced.

## 2. Low Temperature Plasma Control Technology

Plasma regulation is considered a simple, efficient, and low-cost treatment technology [22] that can control the material surface without effects on the surface and matrix of the treated material [23]. The most widely used low temperature plasma generation methods include dielectric barrier discharge, atmospheric pressure plasma jet, glow discharge, radio frequency discharge, etc.

### 2.1. Dielectric Barrier Discharge

Dielectric barrier discharge (DBD) [24] is a kind of non-equilibrium gas discharge with insulating medium inserted into the discharge point space, also known as dielectric barrier corona discharge or silent discharge. Its schematic diagram is shown in Figure 2.

Hossain et al. [25] used tetramethylsilane (TMS) and 3-aminopropyl methyl silane (APDMES) as precursors to deposit superhydrophobic films on glass substrates by dielectric barrier discharge plasma jet. After treatment, the water contact angle is 163°, and the sliding angle is 5°. Lin et al. [26] used CO_2_/N_2_ dielectric barrier discharge plasma and acrylic acid as a precursor to modify the surface of the PTFE film, and then -COOH and other hydrophilic groups (such as C-O, C-O, C-C, etc.) were formed on the surface of PTFE. After two reactions treatments, the water contact angle decreases to 140°, and after twenty reactions, the water contact angle is less than 5°.

Dielectric barrier discharge plasma, due to its high electron energy and fast reaction, can be applied to all kinds of gases without electrode corrosion. Therefore, it has been widely used in sterilization, cleaning material surface, semiconductor manufacturing, and surface treatment.

### 2.2. Atmospheric Pressure Plasma Jet

Atmospheric plasma jet [27] is usually generated by high kilohertz sinusoidal excitation. Through the diagnosis of nanosecond time-resolved discharge images, we can find that it was made up of clumps of fast-moving, high-energy plasma particles, generating one or two discharges in one excitation power cycle. The plasma jet excited by the microsecond high pressure pulse is mainly generated in the rising stage of the high-pressure pulse. Plasma jet characteristics can be regulated by high pressure pulse parameters, which provides a better technical approach for the application of plasma jet. Figure 3 shows how this technique works.

Chen et al. [28] improved the hydrophilicity of the PTFE surface by using an argon plasma jet. After treatment, as the roughness of the root mean square reduced to 5.74 ± 0.32 nm, the surface was smooth and uniform, water contact angle decreased to 28 ± 10°, fluorine content decreased and oxygen content increased. Liu et al. [29] used a nitrogen atmospheric plasma jet to treat the electrochemically corroded aluminum base surface, then the surface was fluorinated with fluoroalkyl silane ethanol. After treatment, the water contact angle decreased to 0°, the fluorine content decreased to 5.67%, and the oxygen content increased rapidly to 46.34%.

Atmospheric plasma jet is suitable for a wide variety of materials due to its low cost, ease of use, and no pollution, which plays an important role in the field of environmental engineering, biomedicine, plasma chemical industry, and so on.

### 2.3. Glow Discharge

Glow discharge [30] (as shown in Figure 4) refers to the gas discharge phenomenon showing glow in low pressure gas, that is, the phenomenon of self-sustaining discharge (self-excited conduction) in rarefied gas.

T. anupriyanka et al. [31] modified the surface of polyethylene terephthalate (PET) fabric with oxygen as process gas in a DC glow discharge plasma chamber. After plasma treatment, the surface energy for water was 3.54 mJ/m^2^. For ethylene glycol, the surface energy was 48 mJ/m^2^. Wang et al. [32] pretreated aluminum alloy after anodic oxidation with low-temperature plasma. Then, nanopores were formed on the surface, and small mastoids were formed at the edge of the pores. Then, the surface was modified by trichlorooctadecylsilane. The carboxyl reaction between trichlorooctyl silane and alumina formed alkyl silane films with low surface energy. Some of the voids form larger microvoids, the cold plasma roughens the surface at the nanoscale and the water contact angle of the surface treated by trichlorooctadecylsilane reaches 157.8°.

Glow discharge plasma can not only treat some material surface and sewage, but also be used as a relatively new display technology due to its advantages of high brightness and fast response.

### 2.4. Radio Frequency Plasma Discharge

Radio frequency low temperature plasma [33] is a low temperature plasma produced by high frequency and high pressure to ionize the air around the electrode, which can produce linear discharge and projectile discharge. Figure 5 shows a schematic of how it works.

Lim et al. [34] treated graphene surface for 20 s with radio frequency plasma of tetrafluoride carbon, after which fluorine was absorbed on the material surface, and formed strong and stable chemical bonds with the surface, making graphene with fluorinated functional groups durable. The water contact angle of the treated surface can be increased to 104.9°. Gursoy et al. [35] deposited poly (butyl hexafluoroacrylate) polymer film on the surface of expanded perlite by radio frequency plasma to successfully prepare the hydrophobic surface. Because of the high fluorine chain structure in butyl hexafluoroacrylate, the deposited polymer has been increased by up to 35.7% fluorine content. The surface which was treated at a power of 20 W can obtain a water contact angle of about 100°.

Radio frequency plasma discharge has high efficiency and produces high chemical active substances, which can give full play to the advantages of efficient chemical treatment technology, but the production efficiency is low in industrial production. Radio frequency plasma has a high chemical treatment efficiency, good performance, and is suitable for the treatment of a small number of samples. Moreover, it will not produce harmful gases and is environmentally friendly, so there is a large demand for radio frequency plasma treatment.

### 2.5. Inductively Coupled Plasma

Inductively coupled plasma is a plasma source that generates current as energy source through electromagnetic induction of a time-varying magnetic field. Its schematic diagram is shown in Figure 6.

Lei et al. [36] used pulse inductively coupled plasma (PICP) to conduct surface modification of polyethylene terephthalate (PET). After modification, oxygen-containing groups were grafted on the surface, and the root mean square roughness increased to 2.525 nm, and the water contact angle reached 38.8°. The treated surface has good biological adhesion. Leet al. [37] used pulsed inductively coupled plasma to treat the surface of polyvinylidene fluoride (PVDF) film. After the treatment, the surface was grafted with alkyl chloride silane, and some spherical particles were distributed on the surface. The water contact angle reached 125.3°. After treatment, it has the effect of treating sewage.

Therefore, inductively coupled plasma processing has obvious advantages in terms of material types and processing speed, and has a large space to play in semiconductor materials and other fields.

## 3. Results and Discussion

### 3.1. Modification Mechanism of Superwetting Surface

The wettability of the solid surface is mainly determined by the corresponding functional groups and the surface topography [38]. In order to achieve superwetting performance, the surface energy must be well calculated and controlled. Chang-hwan Choi et al. [39] established a new model to more accurately estimate the dynamic contact angles of droplets on surfaces with different roughness. For superhydrophobic performance [40], that is, the contact angle (CA) (Figure 7 [41]) value of droplets on the flat hydrophobic surface should exceed 150°, and the surface energy should be less than one quarter of the surface energy of droplets. As a result, most samples with good roughness conditions can achieve 90° hydrophobicity, but it is difficult to achieve 150° super hydrophobicity. In order to achieve ultra-low surface energy, the most commonly used method is to use low surface energy materials or groups to fix Si, F and other particles or elements on the sample surface through spraying [42], deposition, and other processes.

#### 3.1.1. Changes in Surface Chemical Composition

Free energy (or surface tension) on the solid surface can directly affect the wettability and water contact angle of the material surface [43]. The larger the surface free energy is, the easier it is to be wetted. Several common elements that increase surface energy are N, O, Cl, H, F, Si, et al. For example, CF_4_ plasma is used to introduce F-containing functional groups on the surface of SiO_2_ (Figure 8a) [44]; Oxygen low temperature plasma is used to graft O functional groups (Figure 8b) [45] on the glass surface. Therefore, the surface wettability of materials can be regulated by replacing elements on the surface of materials or introducing other functional groups.

Kang et al. [46] studied how the surface wettability can be changed by regulating the generation of hydrophilic groups, such as nitrogen oxides on the surface of polyimide film. After plasma treatment, the water contact angle of the film drops below 30°. Research showed that the peak strength of the carbon–oxygen bond and the carbon–oxygen double bond increased, N/C ratio reached 0.13, and O/C ratio reached 0.45. Peng et al. [47] carried out hydrophilic treatment on the surface of polyethylene glycol by using CF_4_ plasma to graft fluoride onto the surface of polyethylene glycol. The fluorinated layer reduced the interaction between water and polyethylene glycol, resulting in a larger contact angle of 30.7° and the oil contact angle of 60.7°, thus making it easy and effective in anti-pollution and decontamination. Chang-hwan Choi et al. [48] developed a new model to estimate the depinning force of receding droplets on columnar superhydrophobic surfaces with different structures but fixed sizes. This model theoretically proves that the depinning force is linearly related to the maximum three-phase boundary along the droplet boundary, and it is also proved from the side that the hydrophobicity is affected by the surface morphology.

Therefore, the surface of the material can be modified by high-energy or low-energy functional groups to affect the chemical composition of the surface, so as to well control the wettability of the material surface. Some scholars have also tried to graft some polar functional groups on the surface of materials, such as hydroxyl carboxyl groups. The number of polar groups determines the wettability of the surface. However, some materials grafted with chemical functional groups would greatly reduce their working life under different service conditions. For example, the carboxyl group will be destroyed in an alkaline environment, thus reducing its timeliness. Consequently, it is necessary to regulate the microstructure of the surface.

#### 3.1.2. Changes in Surface Roughness

Based on the previous discussion, the chemical composition of smooth surfaces can be regulated and the surface wettability can be controlled by changing the surface free energy, but there are certain limitations. In the Wenzel equation [49], cosθr = r(γ_sv_ − γ_sl_)/γ_lv_, r is roughness, referring to the ratio of the real solid–liquid contact area to the apparent solid–liquid contact area. As known from the equation, when θ < 90°, the contact angle θr of rough surface decreases with the increase of roughness r, and the surface is more hydrophilic; when θ > 90°, θr increases with the increase of surface roughness r, and the hydrophobic surface becomes even more hydrophobic. Using plasma treatment can not only introduce new elements, but also have certain influence on the surface morphology. The schematic diagram is shown in Figure 9a. After treatment, the morphologies were different. The common morphologies included coronal (Figure 9b) [50], dendritic (Figure 9c) [51], globular (Figure 9d) [52], and cauliflower (Figure 9e) [53].

E. Vazirinasab et al. [54] used atmospheric pressure plasma jet to treat the surface of high-temperature vulcanized (HTV) silicone rubber substrate. It is found that the coral-like micron and nanostructures treated by plasma increase the possibility of superhydrophobicity. When the voltage increases, the cluster spacing becomes smaller, the micro-roughness becomes larger, the contact angle of treated water is over than 160°, and the contact angle hysteresis (CAH) is less than 3°. Zhang et al. [55] used micro/nano dielectric barrier discharge to change the wettability of polymethyl methacrylate surface. When the applied voltage is 40 kV and lasts for 300 s, the contact angle increased to 99°. The treated surface is filled with mounds of particles. Results showed that the surface roughness increased to 4.41 nm and root mean square roughness increased to 5.37 nm.

Therefore, when the surface exhibits nano- or sub-micron roughness or surface morphology, it has a very significant effect on the wettability control of the surface. It not only strengthens the timeliness of surface wettability, but also saves the time and cost of repeated surface treatment.

### 3.2. Modification Method of Superwetting Structure on Surface of Different Materials by Plasma

At present, as a simple and efficient surface treatment method, plasma treatment is widely used in all aspects of our life, and has been promoted to process many materials such as cotton fabric [56], fiber fabric [57], and wood [58] material. Plasmas are widely used because they are efficient in plasma processing and deposition, and can treat material surfaces quickly without affecting the overall performance of the material. Besides, the proper temperature required for processing ensures the safety of the production process. In the whole process, no harmful gases are generated, which is friendly to air. This section shows plasma treatment of different materials, as shown in Figure 1.

#### 3.2.1. Textile Fiber Surface

Fibrous materials are made up of many continuous filaments and are used in many fields such as textiles, medicine, architecture, biology, et al., which receive wide attention. Because of the need to contact with rain and dust, textiles with anti-pollution and easy decontaminization properties can improve comfort and reduce washing workload. Fiber surface has a relatively large surface area and high mechanical properties, such as softness, light weight, high strength, air permeability, etc. Therefore, in different application fields, the wettability requirements of material surface are also different. Clothing [59], for example, needs to be hydrophobic and oily. As a result, we need to change the wettability of the fiber surface so to meet the requirements in different conditions. The researches done by other scholars are shown in Table 1.

#### 3.2.2. Porous Material Surface

Porous materials have been widely used in recent years. Advanced porous materials have the advantages of adjustable pore size and specific surface area [66]. For example, porous silicon substrate is widely used in microelectronics, biosensors, and other fields due to its large surface area ratio and certain biocompatibility. Polystyrene foam is widely used in order to prevent liquid from leaking out of food packaging. Improving the hydrophilicity of the surface of the packaging can greatly alleviate this situation and so on. It can be seen that the treatment of porous materials has great application prospects, and some scholars have conducted studies on this, as shown in Table 2.

#### 3.2.3. Porous Material Surface

Polymer films are widely used in chemistry, physics, and certain sectors of biosensors and microelectronic components [72,73]. Using isotropic and anisotropic plasma etching on the polymer substrate surface, ordered nanocolumns or nanoarrays can be obtained to improve hydrophobicity [74]. However, traditional polymer films cannot be widely used due to some defects such as rough surface and poor dry heat performance. For example, due to its low surface free energy and hydrophobic properties, PTFE material is not capable of surface adhesion. Therefore, some scholars modified the surface of polymer film in order to expand its scope of application, as shown in Table 3.

#### 3.2.4. Wood Surface

Due to its high mechanical properties, convenience, and moderate price, wood has become the preferred material for some buildings and decorations [81]. Wood will be subjected to natural degradation, which is an inevitable factor. With the change of time, both microscopic and macroscopic appearance will be changed. The polar groups in the uppermost layer of wood will lead to increased water absorption, expanded wood [82], decomposed microorganisms, increased water content and other problems, thus greatly reducing the service life of firewood. Currently, proposed physical and chemical methods such as impregnation, painting, etc., produce coatings with low adhesion and cannot be used for a long time. Some suggestions have been put forward to prolong the service life of the coatings, as shown in Table 4.

#### 3.2.5. Glass Surface

Due to the advantages of long service life, high light transmittance, and others, glass has been widely used in our daily life and industrial production [87], such as semiconductor, optical equipment, and others. However, due to the disadvantages of high free energy of glass surface, low hydrophobicity, and mechanical instability [88], its application is limited. As a result, some scholars have made some contributions to improve the surface modification of glass, as shown in Table 5.

#### 3.2.6. Particle/Powder Surface

The hydrophobicity of particle surface is very important in many practical applications [92], such as agglomeration and mixing of powder surface, dispersion, and adsorption of particles in composite materials. Some powders have hydrophobicity [93], which makes them float on the liquid surface and unable to form suspension, thus reducing the utilization rate of powders. A superhydrophobic sand has been designed to reduce water evaporation, act as a catalyst, and degrade water pollutants when irradiated by the sun [94]. In recent years, extensive attention has been paid to the modification of wettability of surface [95] such as granular powder. Some scholars have made some studies to increase the application of these granular powder, as shown in Table 6.

## 4. Application of Superwetting Materials

### 4.1. Oil-Water Separation

In industrial production and daily life, a large amount of oily wastewater will be produced, which has a great impact on our life and health. Therefore, separation of oil and water is of great significance. Yan et al. [102] sprayed a hybrid coating of ZnO nanoparticles and waterborne polyurethane on the stainless-steel net. UV light and heat treatment were alternately applied to the treated stainless-steel net to achieve the rapid transformation of superhydrophobicity and superhydrophilicity, so that the mode of oil–water separation is changed from oil removal to water removal. In oil removal mode, the heavy oil in the oil–water mixture can be allowed to permeate; in the water removal mode, the water can be allowed to permeate. The separation efficiency is very high and can be completed only by gravity without external force. In order to separate the oil–water mixture, Liao et al. [103] electrospinned aporous polyvinylidene fluoride–silica composite nanotop layer and a polyvinylidene fluoride intermediate layer on the non-woven fabric carrier. Because the film has high surface energy, it is superhydrophobic in air and oil and under water. Yang et al. [104] constructed long-chain alkyl silane and silica nanoparticles into a hydrophobic surface with micron-scale structure through plasmonic-induced cross-linking. The surface can effectively destroy the stability of oil–water mixture, and the filtration efficiency can reach 99.13%. The purity of the crude oil recovered is up to 99.98%. Kim et al. [105] used atmospheric plasma to process the grid to obtain an underwater ultra-oil-philic functional film layer. The treated grid can separate oil and water at the same time, and the water separation flow rate is high and continuous, and the purity of oil up to 99.99% can be obtained.

The characteristics of the membrane described above can deal with various types of oil–water mixture, and the demand for external driving force is not large. Cross flow filtration can be completed without external driving force, and the purity of the treated oil and water reached more than 99.99%; and the film layer is not easy to be destroyed in the harsh environment of strong acid and alkali. However, in the process of use, oil and other impurities will remain in a large number of voids on the surface of the membrane, which will greatly reduce the membrane flux, membrane life, the hydrophobicity, and the separation efficiency.

### 4.2. Ice Prevention

Ice on solid surfaces can cause serious problems and accidents on roads, power lines, ships, and some energy equipment. There are many traditional methods such as heating, mechanical removal, etc., which require a lot of manpower and material resources. There are also some treatment methods that can produce a series of pollution. Wei et al. [106] found that the prepared superhydrophobic surface can roll down from the surface before the droplet freezed, and the effective contact between the superhydrophobic layer and the droplet was small, which can effectively prevent freezing. Kim et al. [107] covered a smooth and ice-phobic nano-layer on the surface of aluminum. Under the gravity of a low dip angle, both the ice on the surface and unfrozen water can easily flow down, so that the water can drop before freezing, thus reducing the accumulation of water and preventing freezing. Yang [108] et al. used etching as a simple and low-cost method to conduct hydrophobic treatment on the surface of aluminum alloy. The etching solution with the best concentration was selected, and the treated surface obtained a water contact angle of 160°. This hydrophobic surface has significant anti-icing characteristics and can delay the freezing time by 600–700 s. Wang et al. [109] used an aluminate coupling agent to produce hydrophobic surface on aluminum surface. The hydrophobic coating inhibited the growth of frost layer, and the deposition time of frost layer on hydrophobic coating was delayed by 60 min compared with pure aluminum surface. The fatigue test shows that the aluminate coupling agent coating on aluminum surface has stable and strong adhesion ability.

The anti-icing ability of surface decreases with the repetition of icing and melting, and some anti-icing surfaces perform poorly in low temperature and high humidity environment. Therefore, a smooth superwetting layer needs to be deposited in order to prepare a stable superwetting surface that can prevent icing.

### 4.3. Self-Cleaning Antifouling Performance

External environmental pollution will have a serious impact on external surfaces such as glass, ceramics, and metals, making them lose their original characteristics. Piispanen et al. [110] prepared a layer of TiO_2_ coating on the glass surface by a sol-gel treatment process. After ultraviolet irradiation, the TiO_2_ coating turned into a superhydrophobic one, which reduced the adhesion between dust and the glass surface, thus having self-cleaning performance. The copper coating prepared by Yang et al. [111] was easy to be contaminated by oxidation of hydrogen peroxide. The contaminated surface was reoxidized with hydrogen peroxide and then dried in a vacuum at 60 °C for 1 h. The cleaned fabric surface returns to the superhydrophobic state, and the recovered surface has almost the same performance as the uncontaminated surface. D. Anda [112] et al. prepared ultra-liquefied biological coating on the aluminum surface, in which the silica particles were modified by perfluorooctane trioxane. The modified surface was sprinkled with charcoal ash and then poured with water, so that the water droplets on the inclined surface could roll down freely and take away the surface pollutants, which has good self-cleaning performance. J. Lomga et al. [113] used sodium hydroxide and lauric acid to chemically corrode the aluminum surface to prepare the superhydrophobic coating. Figure 10a–c are SEM images etched on aluminum surface with different concentrations of NaOH, Figure 10d shows that the water contact angle on aluminum surface changes with the treatment time. After 200 min, the water contact angle has been more than 160°, which proves that the aluminum surface has reached the superhydrophobic state. Compared with the untreated surface, a few drops of water on the treated surface would wash away the pollutants on the aluminum surface, showing a self-cleaning performance (Figure 10e,f compare the self-cleaning and hydrophobicity of aluminum before and after treatment).

Under the action of the “air cushion effect”, the adhesion of water droplets to the superwetting surface is reduced, and the water droplets will slide down with the surface pollutants, so as to achieve the effect of self-cleaning and pollution prevention.

### 4.4. Biological Field

Superwetting materials are also mostly used in the medical field, and the maintenance and repair of some cells are crucial to the wettability of the material surface. A. O. L. obo et al. [114,115] prepared super hydrophilic vertically aligned carbon nanotubes (VACNTs) for cell adhesion, and the membrane played a great role in the growth of the number of cells during adhesion. In the process of preparation (Figure 11a), the membrane presents a bamboo shape (Figure 11b), and the structure changes (Figure 11d) after plasma etching (Figure 11c). After the treatment, the surface showed a super hydrophilic state (Figure 11f). XPS analysis (Figure 11g) of the surface before and after the treatment showed that the plasma etching efficiency was higher. Carbon nanotubes increased projection on the surface of the cell membrane (Figure 11e), promoted the adhesion and proliferation of the cell membrane, limited the proliferation of cell, and had no adverse reaction on the surface of membrane cells. Lu et al. [116] prepared a superhydrophilic composite film with efficient degradation of tetracycline. The membrane inactivated tetracycline, oxytetracycline, and other biological. In addition, after continuous adsorption and resolution, the film was well preserved, indicating that it had a high utilization rate and stability in the treatment of tetracycline, and the removal rate reached 98.3%. Gorjanc et al. [117] used silver nanoparticles to treat the surface of siloxane with adhesives and then produced hyperaerobic cellulose fibers with strong antibacterial activity against *Escherichia coli* and *Staphylococcus flavans*. Wang et al. [118] grafted zwildes (Carboxylate betaine methacrylate) onto the surface of commercial nanofiltration membrane, which had high anti-adhesion to positively and actively charged proteins. The cleaned membrane had a high recovery rate, and the inactivation efficiency of *E. coli* and *Bacillus Subtilis* can reach 99%. This method can be used for polyamide membrane in all directions. Weng et al. [119] prepared ultra-hydrophilic and antibacterial amphoteric polyamide film composite nanofiltration membrane. The membrane is highly selective to erythromycin and sodium chloride, which can effectively separate the mixture of the two solutions, and the water flow of the prepared film is twice that of the pure film, and has certain antibacterial properties.

The wettability of some nano-biomaterials also affects the adsorption, growth, and survival of some cells and protein fibers. Nanotube materials also play an important role in biomimetic properties and other fields.

## 5. Summary and Outlook

In the past decade, studies using plasma to modulate the surface wettability of various materials have been continuously conducted, which puts forward new views on the use of different plasma generation methods, different precursors, and materials of different sizes.

In this article, we review the influencing factors of wetting surfaces, some of the treatment processes of low-temperature plasmas, and their applications. Although a big breakthrough has been made in the manufacture of wetting surfaces in recent years, more improvements are needed. First of all, the theory and mechanism of super-wetting materials should be further studied, and some super-wetting materials with special structures can be integrated, usually the structure is closely related to the harmony energy, and the integration may give the material new functions. Second, the problem of the binding strength of the prepared wetting film and the matrix; in order to achieve a change in micro-roughness, the thickness of the deposited film should satisfy certain requirements, which makes the process efficiency reduced. At this time, the plasma can be polymerized to deposit a layer of nanoscale composite film which is an ideal method, both to improve durability and to maintain that the corresponding wettability is not affected. Third, to combine the established theoretical models with practical applications, such as the new model mentioned above that measures dynamic water contact angles, whether it can be applied to most wetting materials is worth following up.

## Figures and Tables

**Figure 1 polymers-14-03759-f001:**
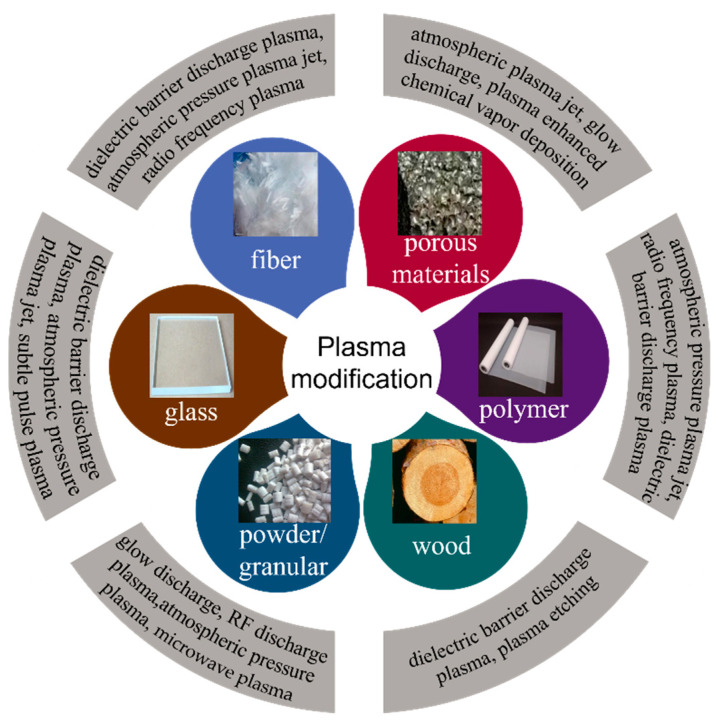
Schematic diagram of plasma treatment of different materials.

**Figure 2 polymers-14-03759-f002:**
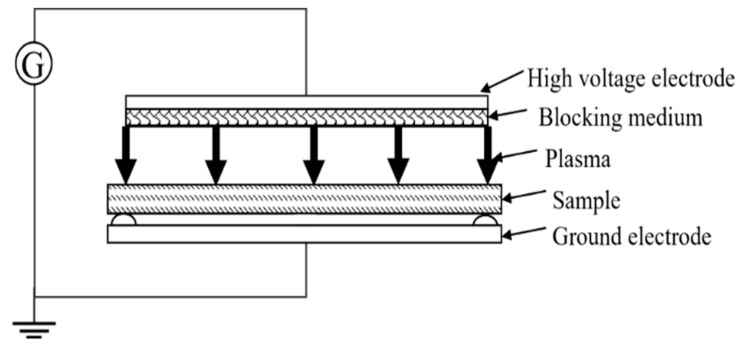
Dielectric barrier discharge plasma schematic diagram.

**Figure 3 polymers-14-03759-f003:**
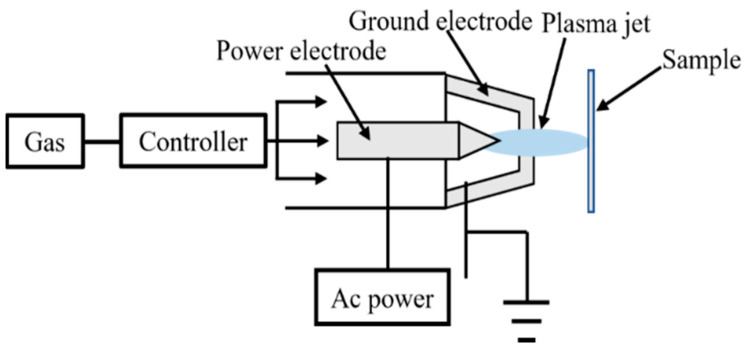
Schematic diagram of atmospheric plasma jet.

**Figure 4 polymers-14-03759-f004:**
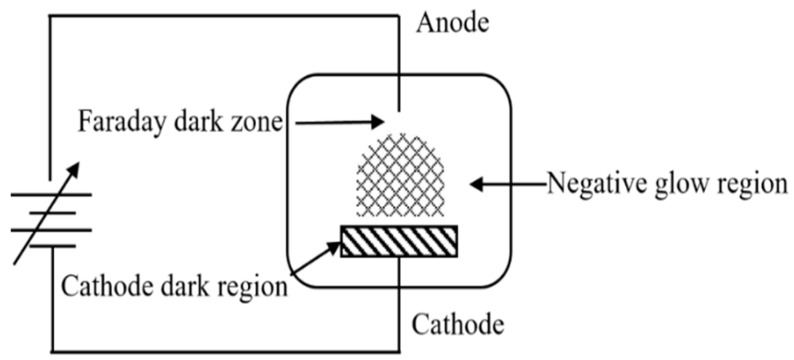
Schematic diagram of a low pressure glow discharge.

**Figure 5 polymers-14-03759-f005:**
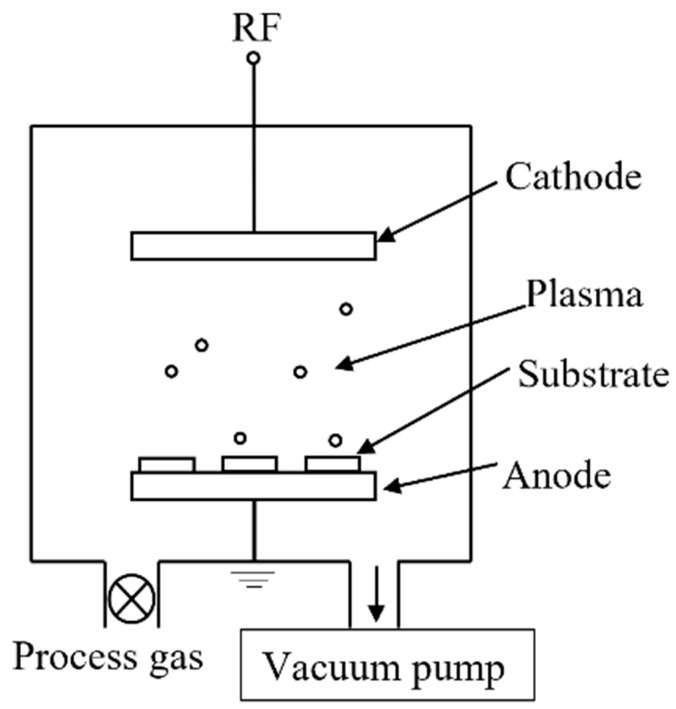
Working principle diagram of DC radio frequency plasma.

**Figure 6 polymers-14-03759-f006:**
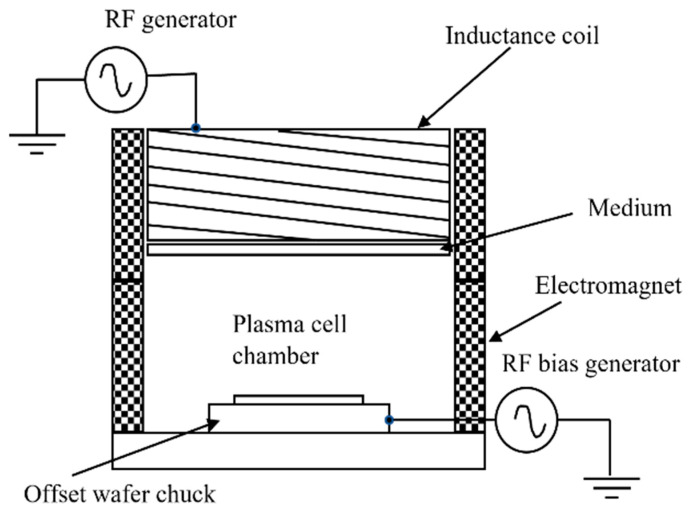
Schematic diagram of inductively coupled plasma.

**Figure 7 polymers-14-03759-f007:**
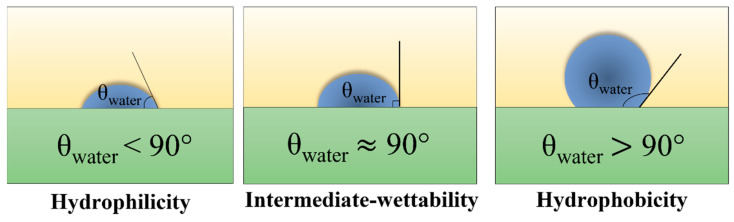
Diagram of water contact angle measurement [41].

**Figure 8 polymers-14-03759-f008:**
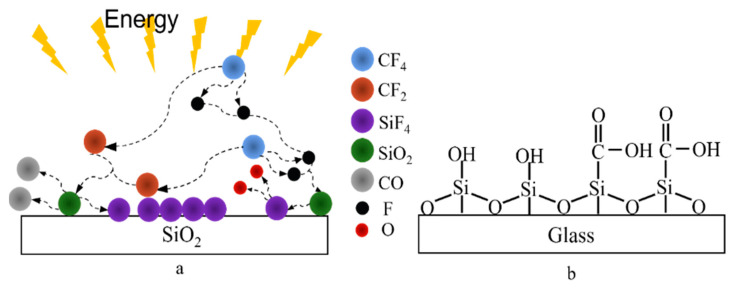
(**a**) Introducing F-containing functional groups into the surface of SiO_2_ treated by plasma; (**b**) plasma-treated glass surface grafted with O functional groups.

**Figure 9 polymers-14-03759-f009:**
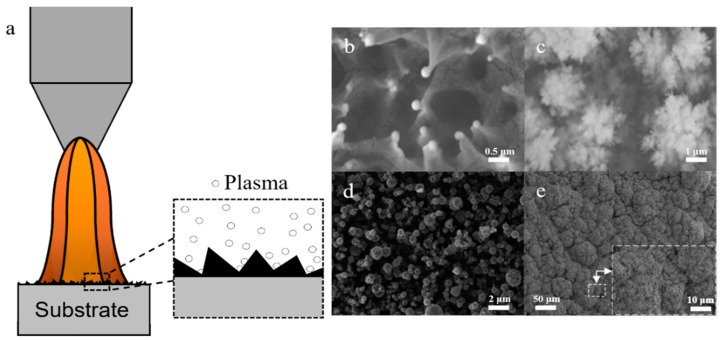
(**a**). Principle of plasma treatment; (**b**) coronary [50]; (**c**) dendritic [51]; (**d**) ball [52]; (**e**) cauliflower [53].

**Figure 10 polymers-14-03759-f010:**
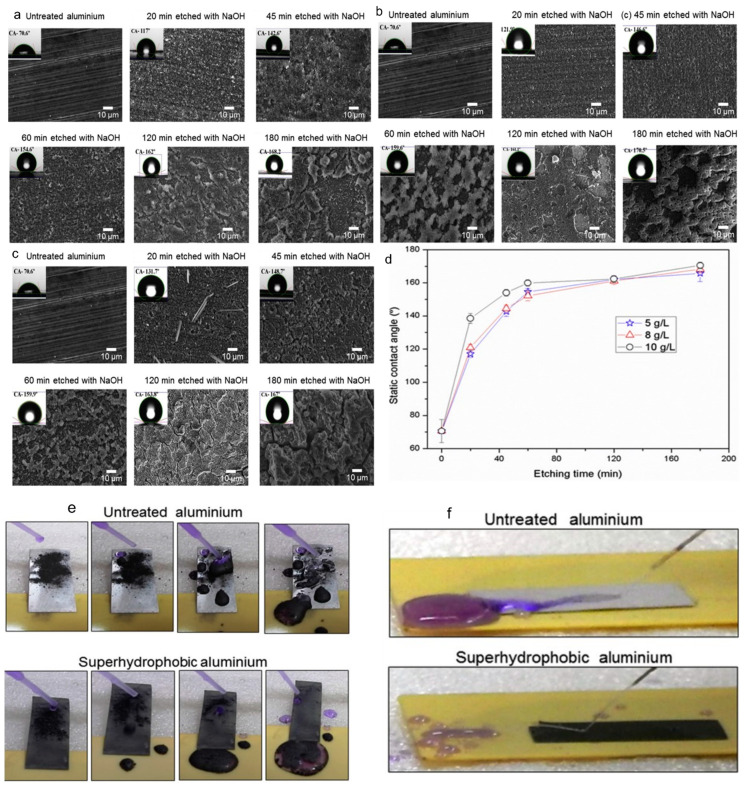
(**a**) The aluminum surface was etched with 5 g/L NaOH for different times; (**b**) the aluminum surface was etched with 8 g/L NaOH for different times; (**c**) the aluminum surface was etched with 10 g/L NaOH for different times; (**d**) static contact angle versus etching time for different NaOH concentration. (**e**,**f**) comparison of self-cleaning and hydrophobicity of aluminum before and after treatment [113].

**Figure 11 polymers-14-03759-f011:**
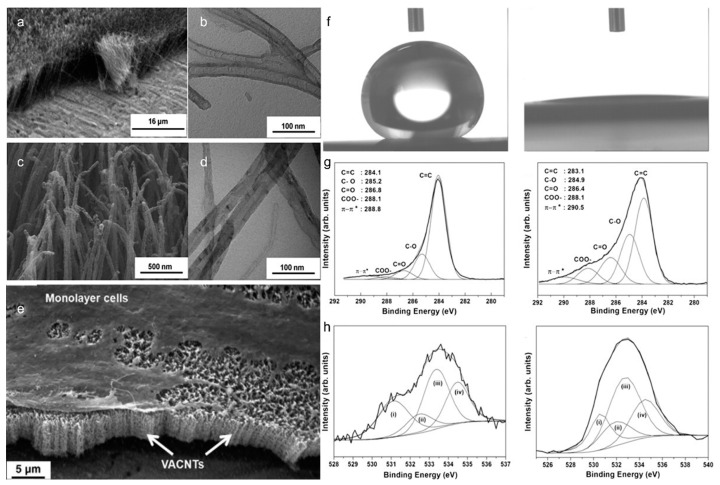
SEM images of (**a**) as-grown films and (**c**) superhydrophilic VACNT films; TEM images of (**b**) as-grown films and (**d**) superhydrophilic VACNT films; (**e**) characterization of human chondrocytes on VACNT films; (**f**) change of water contact angle before and after treatment from 154° to 0°; (**g**) C1s XPS peak analysis before and after; (**h**) O1s XPS peak analysis before and after (π-π*: The unsaturated compound transitions from the ground state (π) to the excited state (π*)) [114].

**Table 1 polymers-14-03759-t001:** Plasma treated fiber surface.

Material	Gas	Plasma Type	Working Conditions	Involves Elements/ Functional Groups	Morphology Change	Water Contact Angle	Ref.
Cellulose	C_2_F_5_H/Ar	RF	120 °C, 1 Torr, 30 W, 13.56 MHz	F	No change	0~104°	[60]
Cotton	HDTMS/N_2_	Glow discharge	3 × 10^−3^ mbar, 400 W, 13.56 MHz	-COO^−^, O-O^−^	Submicron particle	148.2~161.3°	[61]
Cotton	n-Hexane/Ar	APPJ	V_(DC)_ = 20 kv	O	Grain size larger	0~154.7°	[62]
Faceted nonwovens	HMDSO/Ar	APP	101 kPa, 300 V, 19 kHz	Si-C, Si-CH_3_	NPs	155°	[63]
Nanofibre pad	N_2_/He	DBD	101 kPa, 9 W	N	No change	57 ± 0.2°~ 10.6 ± 2°	[64]
Cellulose	N_2_/NH_3_	DBD	25 °C, 101 kPa, 1000 W	-NH_2_, -CONH_2_	Roughness increase	40~22°	[65]

(HDTMS: Hexadecyltrimethoxysilane; HMDSO: Hexamethyldisiloxane; C_2_F_5_H: pentafluoroethane).

**Table 2 polymers-14-03759-t002:** Plasma treatment of porous materials.

Material	Gas	Plasma Type	Working Conditions	Involves Elements/ Functional Groups	Morphology Change	Water Contact Angle	Ref.
Porous aluminum	HMDSO/H_2_	APPJ	101.3 kPa, 25 °C, 15–25 kHz	Si-O-Si	Dendritic	<90°~ >150°	[51,67]
Porous aluminum	OTS/ Air	Glow discharge	30–50 kPa, 150–250 W	Si, Cl	Nanopore size	157.8°	[32]
Polystyrenefoam	O_2_	Plasma treatment	240 W, 40 kHz, 0.12 mbar	C=O, O-C=O, O-C(=O)-O	/	86.0~15.13°	[68]
Porous fiber	Air	APP	101 Kpa	-COOH, -OH	NPs	81.3~72.5°	[69]
Porous silicon	O_2_	Plasma treatment	1 Torr, 10.5 W	Si-OH	No change	64.5°~<0.5°	[70]
Filter paper	HMDSO/n-Hexane	PECVD	80 W, 500 mToor	C-H_n_	Double membrane	0~141.5°	[71]

(OTS: Octadecyltrichlorosilane; HMDSO: Hexamethyldisiloxane).

**Table 3 polymers-14-03759-t003:** Plasma surface treatment of polymer.

Material	Gas	Plasma Type	Working Conditions	Involves Elements/ Functional Groups	Morphology Change	Water Contact Angle	Ref.
Polyethylene film	O_2_	RF	90 °C, 1 Torr, 200 W	C=O, C-O	Nanostructures	97.2~152.9°	[75]
Polyimide film	He	DBD	80 kHz, 1.5 kV	C-O, C=O	Roughness increase	>70°~<30°	[46]
PTFE	O_2_	RF	6.66 Pa, 13.56 MHz, 20–70 W	O	Alveolar structure	110~152.8°	[76]
PTFE	O_2_/Ar	RF	150 W,3 h	/	Coronary structure	110~178.9°	[50]
PTFE	Ar	APPJ	4.4 kV,1.1 W	O	Roughness reduction	100~28°	[28]
PTFE	Ar	APPJ	9.6 kHz, 10 ± 2 °C, 26 kV	^·^OH	Irregular bulge	109~37°	[77]
Polyurethane foam	Aerosol/He	DBD	22 kHz, 10–60 min	O	Spherical particle	155 ± 5°~ 80 ± 2°	[52]
PTFE	CO_2_/N_2_	DBD	6.2 W, 19.4 kHz	C=O	Surface compactness	140.9~48.6°	[26]
PTFE	O_2_	CCP	13.56 MHz, <70 s °C, 13 Pa		Roughness increase	\	[78]
PTFE	O_2_	CCP	13.56 MHz, 13 Pa	-CF_2_	Roughness increase	\	[79]
Ethylene propylene	O_2_, C_4_F_8_	ICP, RIE	1900 W, 0.75 Pa; 400 W, 10 mT	F	Roughness increase	95~168°	[80]

(PTFE: Polytetrafluoroethylene; RIE: Reactive Ion Etcher).

**Table 4 polymers-14-03759-t004:** Plasma treatment of wood surface.

Material	Gas	Plasma Type	Working Conditions	Involves Elements/ Functional Groups	Morphology Change	Water Contact Angle	Ref.
Chinese fir	HMDSO/Ar	DBD	95 kHz, 10 kV	Si-C; Si-O-Si	Roughness reduction	<1°~137.7°	[83]
Wood	N_2_/N_2_O/Vapor crystallization solution of ZnO NPs	DBD	1 bar, <50 °C	Si-OH	Spherical particle	40~100°	[84]
Wood	O_2_	RF	13.56 MHz, 0.5 Toor	C=C, C-C	Roughness increase	0~153°	[85]
Poplar	HMDSO	DBD	60 W, 20 KPa, 75 s	Si-O-Si, Si-O-C	Acicular structure	81~127.7°	[86]

**Table 5 polymers-14-03759-t005:** Plasma treatment of glass surface.

Material	Gas	Plasma Type	Working Conditions	Involves Elements/ Functional Groups	Morphology Change	Water Contact Angle	Ref.
Organic glass	Air	DBD	250 MHz, 30 kV	-CH_3_	No change	71~92°	[89]
Organic glass	CF_4_	DBD	1 × 10^5^ Pa, 40 kV, 500 Hz, 2.25 W	F	Roughness increase	68~99°	[55]
Glass	Ar/TMS	APPJ	22 kHz	Si-O, Si-C	Uniformly convex	67~110.3°	[90]
Slide	Aerosol/He	DBD	22 kHz, 10–60 min	-COO, C-O	Spherical particle	160~<5°	[52]
Organic glass	He	APPJ	16 kV, 1 kHz	O=C-O, C-OH	Small peak	27°	[91]
Glass	O_2_	Low temperature plasma	298 K	-OH, COOH	Roughness increase	21.1~2.6°	[45]

(TMS: Trimethylsilanol).

**Table 6 polymers-14-03759-t006:** Plasma treatment of particle surface.

Material	Gas	Plasma Type	Working Conditions	Involves Elements/ Functional Groups	Morphology Change	Water Contact Angle	Ref.
Clubmosses	O_2_/N_2_	ICP	10 MHz, 20 W, 0.8–40 Pa	O	No change	140~60°	[96]
SiO_2_	ppOD	ICP	13.56 MHz, 0.7 Pa, 4–80 W	-CH^−^	Spherical particle	37~>90°	[97]
Al_2_O_3_	CO_2_/H_2_	Plasma deposition	13.56 MHz, 80 Pa	-OH, COOH	Clusters particles	46~74°	[98]
PMMA	CF_4_	RF	10 Pa, 250 kHz	F	Small hills	115.6~150.6°	[99]
PP	He	APP	20 KHz, <28.9 °C	O-CO-O	/	99~69°	[100]
Artificial sandstone	Air	microwave plasma	25 °C, 1 kHz, 500 W	Si-OH	/	123.6~58.5°	[101]

(PP: Polypropylene).

## Data Availability

Not applicable.

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
