# Peer review of "A Review on Research Progress in Plasma-Controlled Superwetting Surface Structure and Properties"

_polymers, 2022, doi:10.3390/polym14183759_

Round 1

Reviewer 1 Report

I read the mini review paper about Plasma-Controlled Super-wetting Surface Structure and Properties by Li et al. 

The topic is highly interesting but the content of the paper is limited and it does not reflect the current state of the art in the field according to my opinion. Thus, in order to be published I suggest major revision.

In particular, 

1. Literature should be significantly improved by adding more papers reporting wetting control using plasma technologies. Some examples:

A.    Papers from the group of Chang-Hwan Choi - Stevens Institute of Technology

B.    Papers from Rosa di mundo et al. and the university of Bari group about plasma deposition

C.     Papers from NCSR Demokritos plasma group about plasma micro-nanotexturing & deposition and their applications (i.e. Ellinas et al. Ultra-low friction, superhydrophobic, plasma micro-nanotextured fluorinated ethylene propylene (FEP) surfaces, Micro and Nano Engineering 14, 100104, and many more)

2. The sections about the properties of such surfaces require extensive revision to include more example of plasma engineered surfaces used in the applications mentioned by the authors. The addition of a representative figure would also improve the content.  

3. References are not displayed correctly in the text. In some cases are shown as one number i.e. line 31 references 10 and 11 are shown as 1011. Please use brackets and commas to separate one from another.

4. I really like figure 7 and I believe it should be mentioned earlier (i.e. in the introduction section) to visualize which materials can be processed with different plasma technologies.

5. In the summary section, I believe that the authors should discuss what has been done up to now in order to improve the durability of superwetting surfaces (there are several review and research papers in the area) and what else should be done according to their point of view in order to further improve it and not just reporting the well-known problem of durability of such surfaces.

Reviewer 2 Report

The review paper of Li et al. reports on the plasma technologies that control the surface superwettability. They focused the review in the surface structure/properties of different materials and, finally, described the applications of the superwetting materials. In general, the paper is well-structured and the subject is of interest for the readers of Polymers MDPI journal.

There are several typos along the text that needs attention by the authors. Moreover, in oder to improve paper quality, some points are addressed:

(1) Please verify the citation along the paper text, there are only numbers without the conventional citation structure [ ... ];

(2) It is interesting to present in the introductory topic the temporal evolution of the number of papers published on plasma technologies that control the surface superwettability, evidencing for the reader the crescent motivation in this research area. The database of Scopus and Web of Science can be used;

(3) Page 3, line 111: The corret is only "Glow discharge" or "Plasma". Please correct along paper text;

(4) Figure 3: Please substitute "Working principle diagram of glow discharge plasma." by "Schematic diagram of a low pressure glow discharge". 

(5) Figure 4: Please substitute "Working principle diagram of DC radio frequency plasma." by "Schematic diagram of a radio frequency plasma operated at low pressure". Please modify "-V(DC)" in the figure 4 by "RF".

(6) In addition, it is necessary to improve the text of the topic 2.4, explaining for example the PECVD process;

(7) It is interesting to insert a topic 2.5 to discuss the ICP reactor (and plasma etching process) that is cited in several plasma treatment in topic 3;

(8) Figure 7: Please explain the "subtlepulse plasma", it is not observed citation of this technique along the paper text.

Round 2

Reviewer 1 Report

My major comments were addressed. The paper can be now accepted for publication.

Reviewer 2 Report

The authors performed the requested corrections very well, it is recommended to publish the review article in this form.